# Virus infections in honeybee colonies naturally surviving ectoparasitic mite vectors

**Melissa A. Y. Oddie**[1]*, **Sandra Lanz**[2], **Bjørn Dahle**[1], **Orlando Yañez**[2], **Peter Neumann**[2]

**1** Norwegian Beekeepers Association, Dyrskuev, Kløfta, Norway, **2** Institute of Bee Health, Vetsuisse Faculty, University of Bern, Bern, Switzerland

* melissa.oddie@norbi.no

**Data Availability Statement:** All relevant data are within the manuscript and its Supporting Information files.

**Funding:** Financial support was granted to Peter Neumann and Bjørn Dahle by the Vinetum

## Abstract

Western honeybee populations, *Apis mellifera*, in Europe have been known to survive infestations of the ectoparasitic mite, *Varroa destructor*, by means of natural selection. Proposed mechanisms in literature have been focused on the management of this parasite, however literature remains scare on the differences in viral ecology between colonies that have adapted to *V. destructor* and those that are consistently treated for it. Samples were collected from both a mite-surviving and a sympatric mite-susceptible honeybee population in Norway. The prevalence and abundances of 10 viruses, vectored by the parasite or not, were investigated in adult host workers and pupae as well as in *V. destructor* mites. Here we show that the mite-vectored Deformed wing virus (DWV-A) is often lower in both abundance and prevalence in the mite-surviving population in tandem with lower phoretic mite infestations compared to the mite susceptible population. However, the non-mite-vectored Black queen cell virus (BQCV), had both a higher abundance and prevalence in the mite-surviving population compared to the susceptible population. The data therefore suggest that general adaptations to virus infections may be unlikely to explain colony survival. Instead, mechanisms suppressing mite reproduction and therefore the impact seem to be more important.

## Introduction

Honeybee pathogens like Deformed wing virus (DWV) and its variants, which in the past were largely benign [1], have been given novel transmission pathways through the invasive ectoparasitic mite vector *Varroa destructor* [2]. The mite bolsters viral titres to the point of reducing the longevity of individual bees [3]. This leads to colony weakening and collapse in many cases [3, 4]. As a result, *V. destructor* is one of the most serious threats to Western honeybees, both managed and wild [3–5]. Due to the limited defence mechanisms of Western honeybees, unhindered *V. destructor* population growth can cause numbers to reach fatal levels in late summer and autumn in temperate climates, when the bees being reared are required for winter colony hibernation [4, 6]. There is now very solid evidence that Western honeybee populations, if left untreated, have the potential to adapt to *V. destructor* infestations without the need for human-mediated mite control [7–9] (via large initial colony losses), and that they can develop this ability in as little as five years [10]. Previous literature has pointed to several

foundation and by the Research Council of Norway (grant. Nb 207694) respectively. The funders had no role in study design, data collection and analysis, decision to publish, or preparation of the manuscript.

**Competing interests:** The authors have declared that no competing interests exist.

mechanisms such as reduced post-capping period [10], potential changes in brood volatiles [11], grooming [12], brood removal [13, 14] and brood cell recapping [15]. One of the most prominent traits that has been detected in naturally surviving honeybees, regardless of the mechanisms identified, is suppressed mite reproduction (SMR) [7, 16], signified by lower average reproductive output per foundress mite. SMR has been recorded not only in Africanized bee populations surviving *V. destructor* [9, 17], but in the "more susceptible" European populations as well, including the Primorsky bees from Russia [18], the Gotland bees in Sweden, the Avignon bees in France [16, 19], in England, Cuba [9] and a population in the Oslo region of Norway [8].

The managed population of bees in Norway able to persist without mite treatments (mite-surviving) [8] has been used commercially since before the introduction of *V. destructor* into the local area approximately 30 years ago [8]. Selection efforts employed by the beekeeper managing the population included only the monitoring of docility and high honey productivity, and these traits were preserved along with the development of the mite-surviving adaptations due to natural selection (The removal of all treatment for *V. destructor* and subsequent loss of colonies without adaptive traits) [8]. Evidence has been gathered that suggest these bees possess mechanisms that focus on controlling the parasite directly, through SMR [7, 8, 15], however there has been no study monitoring the loads of key viruses in the system. It is possible that observed differences between the viral profiles of this population and a local control may provide more insight into the effect these "mite surviving" traits have on viruses.

The Gotland population of Varroa-surviving bees did display evidence of viral tolerance in a laboratory study, by retaining comparative levels of tested viruses but presenting a lower mortality rate [20]. If parasite suppression plays a central role in the ability of honeybees to persist through infestations, we can expect associated virus prevalence and abundance to be lower when the vector levels are also low. Levels of non-vectored viruses should remain relatively similar. Though observing similar levels of Varroa-vectored viruses between both groups will not be sufficient evidence to suggest viral tolerance, the possibility cannot be excluded, and it will provide further insight into the dynamic.

Here, we measured the abundance and prevalence of mite-vectored, non-mite-vectored and mite-associated viruses in parallel with the vector (*V. destructor*) infestation levels in the Norwegian mite-surviving honeybee population and compared the information to a sympatric, regularly treated control population (mite-susceptible). Viruses were measured throughout the active season in host workers. These viruses were: those known to be vectored by *V. destructor*: Deformed wing virus A & B, Slow bee paralysis virus, Israeli acute paralysis virus, Kashmir bee virus and Acute bee paralysis virus (DWV-A, DWV-B, SBPV, IAPV, KBV and ABPV), viruses not known to be vectored: Black queen cell virus and Lake sinai virus (BQCV and LSV), and viruses known to be associated with but not directly transmitted by *V. destructor*: Sacbrood virus and Chronic bee paralysis virus (SBV and CBPV). DWV-A, DWV-B and SBPV were also monitored in both developing pupae and the mites contained in the same brood cells. Colony-level *V. destructor* infestation levels were measured during the virus sampling period.

## Materials and methods

### Sample collection and *Varroa destructor* infestation rates

In September 2013, April and June 2014, samples were taken in Østlandet (South-eastern Norway) from local queenright *A. mellifera* "Buckfast" colonies known to survive *V. destructor* infestation without treatments for at least 16 years [21] (n = 32, 3 apiaries). Samples were also

collected from colonies regularly treated against these mites with oxalic acid and/or drone brood removal (n = 69, 7 apiaries). Control colonies were managed by several beekeepers separately from the surviving colonies. All colonies were within 100km and placed in similar farmland habitat. Management practices were focused on honey production, centred around a single flowering crop (raspberry flower) and feeding schedules were designed around this, accounting for local variation in flowering times. Adult workers were collected from outer frames of the brood box. Phoretic mites were sampled using routine washing methods (~100–400 bees, [22]). Infested pupae were sampled, and their associated mites collected and stored. All samples were transported on ice to Bern, Switzerland [23] and then stored at -80˚C until processing.

## Sample selection and analytic approach

**Pooled samples.** One hundred workers and phoretic mites of the colonies sampled in April (spring) and June (summer) 2014 (n = 58; surviving = 24, susceptible = 34) were pooled and homogenized for each colony. Virus prevalence and abundance were analysed using PCR and qPCR techniques respectively.

**Individual samples.** Adult workers (n = 11–13 per colony) and phoretic mites (1–22 per colony) were sampled from 10 surviving and 11 susceptible colonies in September (autumn) 2013 and analysed for the presence of DWV-A. Infested honeybee pupae were collected with their associated mites from 47 colonies at eight apiaries in all seasons (29 colonies from three surviving apiaries and 18 colonies from five susceptible ones). Brood samples were taken in April and June from 15 surviving and 19 susceptible colonies. All pupae and mites preserved for viral analysis were processed by individual for DWV-A, DWV-B and SBPV.

**Homogenization and RNA extraction.** TN buffer (Tris 10mM, NaCl 10 mM; pH 7.6) was added to each sample (25 ml for pooled workers, 100–300 μl for pooled phoretic mites, 250 μl for individual workers and pupae and 100 μl for individual mites) and the sample was homogenized with either a Dispomix® Drive homogenizer (Medic tools) for pooled worker samples or a tissuelyser (Qiagen Retsch MM300, 1 min at 25g/s) for pooled phoretic mites and all individual samples [24]. An aliquot of 50 μl homogenate was used for RNA extraction using the NucleoSpin® RNA II kit, (Macherey-Nagel) following the manufacture's recommendations. The total extracted RNA was diluted in 60 μl of RNase-free water.

**Reverse transcription, PCR and qPCR assays.** The RNA was transcribed to cDNA using M-MLV reverse transcription kit (Promega) following the manufacturer's recommendations using a defined amount of RNA (1μg for bees and 50 ng for mites, respectively) according to fluorospectrometry (NanoDrop[TM] 1000) measurements [24]. cDNAs were diluted 10-fold in nuclease-free water. With a standard qualitative PCR (0.125 My Taq[TM] polymerase (Bioline), 5μl 5x buffer, 1 μl of the respective forward and reverse primers (S1 Table) and RNase-free water to complete 25 μl final volume; 2 min at 95˚C, 35 cycles with 20 sec at 95˚C, 20 sec at 57˚C and 30 sec at 72˚C, 2 min at 72˚C), pooled worker and phoretic mite samples were then screened for the following viruses using routine protocols [25]): DWV-A, DWV-B, ABPV, IAPV, KBV, CBPV, SBPV, SBV, BQCV, LSV1 and LSV2. Similarly, pooled brood samples were screened for DWV-A, DWV-B, ABPV, IAPV, KBV, SBV, LSV1, LSV2, SBPV and CBPV. Positive and negative controls were used for each PCR run. Each PCR Product was analysed on 1.2% agarose gel. The agarose gel was stained with GelRed[TM] and visualized by UV light. With quantitative RT-PCR (RT-qPCR; Kapa SYBR® Fast Master Mix (KAPA, Biosystems), 10 μl master mix, 3 ul of the 1:10 diluted cDNA template, 0.4 μl forward and reverse target primers (10 mM) and 6.2 μl RNase-free water; 3 min at 95˚C, 40 cycles of 95˚C for 3 sec and 55˚C for 30 sec; melting curve: 55–95 ˚C with 0.5˚C intervals s$^{-1}$) pooled worker samples,

where viruses were detected with qualitative PCR, were analysed to determine virus levels (DWV-A, BQCV, LSV1, LSV2 and SBPV). Individual adult workers and phoretic mites were analysed individually for DWV-A by use of qPCR (protocol described above). Individual brood samples (pupae and brood mites) were analysed individually for the viruses detected in the PCR (DWV-A, DWV-B, and SBPV, protocol described above). Standard curves prepared from viral and β-Actin gene targets [26] were used for virus quantification and normalization, respectively. The standard curves were established by plotting the logarithm of 10-fold dilutions of purified PCR products ($10^{-3}$ to $10^{-6}$ ng/reaction) against the corresponding Ct value as the average of two repetitions. The PCR efficiency (E = 10(−1/slope) − 1) and the linear standard equations for each target were as follows: E = 95,5%, Ct = −3.434 × +47.762, $R^2$ = 0.999 for DWV-A; E = 101,9%, Ct = −3.275× +30.881, $R^2$ = 0.998 for BQCV; E = 96,3%, Ct = −3.415 × +37.875, $R^2$ = 0.988 for LSV-1; E = 85,5%, Ct = −3.724 × +36.203, $R^2$ = 0.971 for LSV-2; E = 98,5%, Ct = −3.358 × +32.064, $R^2$ = 0.999 for SBPV and E = 110,4%, Ct = − 3.094 × + 33.163, $R^2$ = 0.994 for β-Actin. Resulting viral loads for the respective amount of RNA included in the reverse transcription reaction (1μg for bees and 50 ng for mites) were then adjusted by the various experimental dilution factors to account for the total volume of RNA per sample. A Cq cut-off value (according to the value of the negative control) was used to define the target status (positive or negative).

**Sequencing.**    To confirm the virus identity, selected PCR-products were purified using the NucleoSpin® Gel and PCR Clean-up kit (Macherey-Nagel) following the manufacturer's recommendations. The purified amplicons were commercially sequenced (Fasteris SA) and compared with reference sequences deposited in GenBank.

## Statistical analyses

Statistical analyses were performed using the R statistical computing software (version 4.1.2) [27] and the packages LME4 [28], gamlss [29] and nlme [30]. Phoretic mite infestation rate data were analysed using generalized linear models with a Poisson error structure, including season and population type (susceptible (treated) or surviving (untreated)) as fixed effects. Worker brood infestation rate proportions were compared using a zero-inflated beta regression model. Virus prevalence data extracted from infested pupae were analysed using generalized linear models with a binomial error structure including season, population type, pupal age, and species (mite or bee) as fixed effects. Individual worker samples were only tested for DWV-A and prevalence was analysed using a $\chi^2$ test. Viral abundance data were log transformed and analysed using linear regression models with population type, brood and adult bee infestation rates, and virus as fixed effects. A post hoc analysis was included investigating the relationship of each virus to the other parameters and a Bonferroni correction was applied to account for multiple testing (α = 0.025).

## Results

### Mite infestation levels

Mite-surviving colonies had significantly lower phoretic mite infestation levels than susceptible colonies when all respective apiaries were pooled ($\chi^2$ 0 5.88, df = 1, p = 0.017, Fig 1 and Table in S1 Table). Mite infestation levels were not significantly affected by the time of year (fall 2013, spring 2014 or summer 2014, Table in S1 Table) when all samples were considered together. Brood infestation rates were not significantly different between surviving or susceptible populations in spring and summer (Table in S1 Table); however, no data were collected on brood infestation rates in autumn (Table in S1 Table).

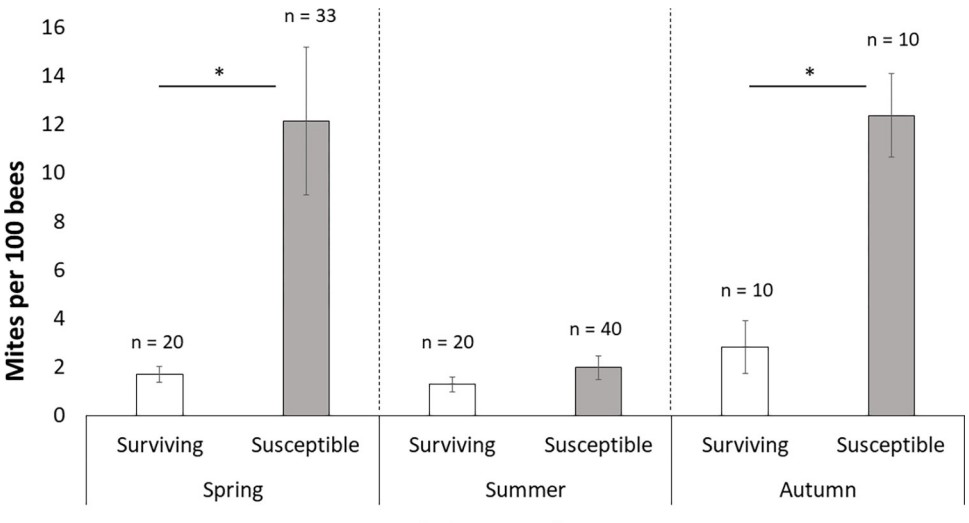

**Fig 1. Number of mites, *Varroa destructor*, per 100 adult workers per colony in both mite-surviving and mite-susceptible host colonies, *Apis mellifera*, in spring, summer and autumn.** Susceptible colonies had significantly more mites in spring and autumn ($\chi^2$ 0 5.88, df = 1, p = 0.017), standard errors shown.

## Virus prevalence and abundance

**Pooled adult worker and mite samples.** In the pooled worker and mite samples from summer 2014, the following viruses were detected: BQCV, DWV-A, LSV1, LSV2 and SBPV. Sequencing of the PCR products confirmed their identity (GenBank accession: BQCV, EF517515.1, 482 bp query length, 100% query cover, 98,55% identity; DWV-A, JF346624.1, 369 bp query length, 100% query cover, 98,37% identity; LSV2, KY465710.1, 116 bp query length, 100% query cover, 97,41% identity; SBPV, NC_014137.1, 167 bp query length, 100% query cover, 100% identity). The LSV sequence produced with the LSV1 primers was more similar to, and therefore recalled as, LSV8 (Schroeder et al. 2022, ON108639.1, 63 bp query length, 88% query cover, 89,29% identity) [31].

*Virus prevalence.* The surviving colonies had a significantly lower prevalence of DWV-A ($\chi^2$ = 4.61, df = 1, p = 0.03, Table in S1 Table), but a higher prevalence of BQCV than the susceptible colonies ($\chi^2$ = 15.03, df = 1, p<0.001, Fig 2 and Table in S1 Table). None of the other viruses showed significantly different rates between the two population types. DWV-A was generally more common in mites than in tested adult workers ($\chi^2$ = 8.7, df = 1, p = 0.003, Fig 2 and Table in S1 Table) and prevalence of DWV-A did vary with season ($\chi^2$ = 28.64, df = 2, p < 0.001, Table in S1 Table).

*Virus abundance.* When comparing virus abundances between pooled worker samples from surviving and susceptible colonies, the surviving colonies had higher BQCV loads (F = 39.64, df = 1, p < 0.001, Fig 3 and Table in S1 Table), but no significant differences were detected for DWV-A, LSV1, LSV2 and SBPV (Table in S1 Table). In the pooled adult samples, DWV-A titres did increase significantly with the proportion of infested brood in a colony (F = 8.18, df = 1, p = 0.01, Table in S1 Table). Abundance of LSV1 was significantly reduced with an increase in mite brood infestation levels (F = 55.56, df = 1, p <0.001, Table in S1 Table).

**Individual adult worker samples and phoretic foundress mites.** *Prevalence (DWV-A only).* Viral prevalence of DWV-A in individual adult worker samples was lower in surviving colonies ($\chi^2$ = 33.751, df = 1, p < 0.01). However, mites showed no difference in viral prevalence between the population types (surviving or susceptible colonies, Fig 4).

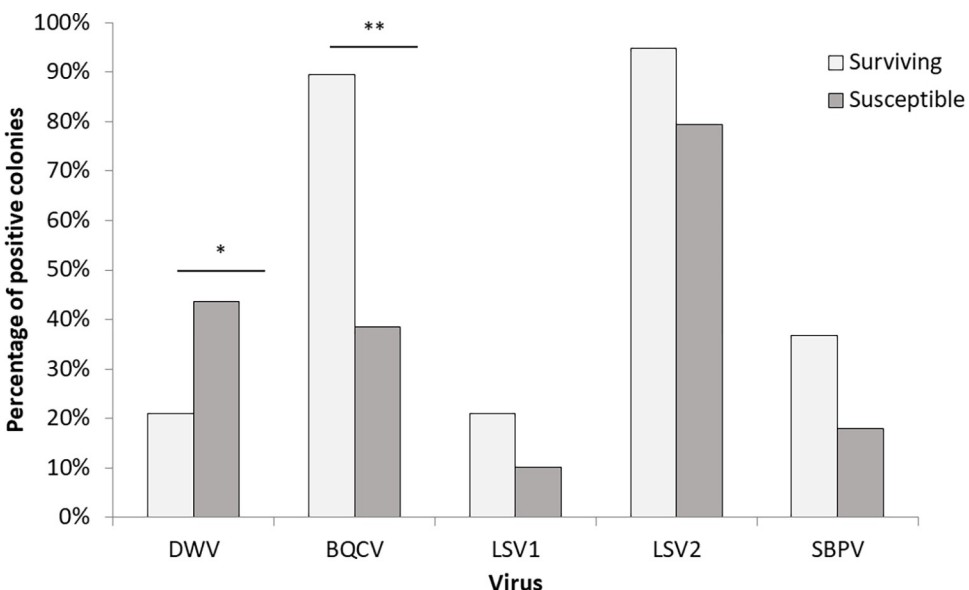

**Fig 2. Colony-level prevalence of the detected viruses in pooled adult worker samples in June 2014.** Surviving (untreated) colonies had a significantly higher BQCV prevalence ($\chi^2$ = 15.03, df = 1, p<0.001, Table in S1 Table), but a lower DWV-A prevalence ($\chi^2$ = 6.75, df = 1, p = 0.009, Table in S1 Table), when compared to susceptible (treated) colonies. Sampled colonies n = 31 susceptible and n = 19 surviving.

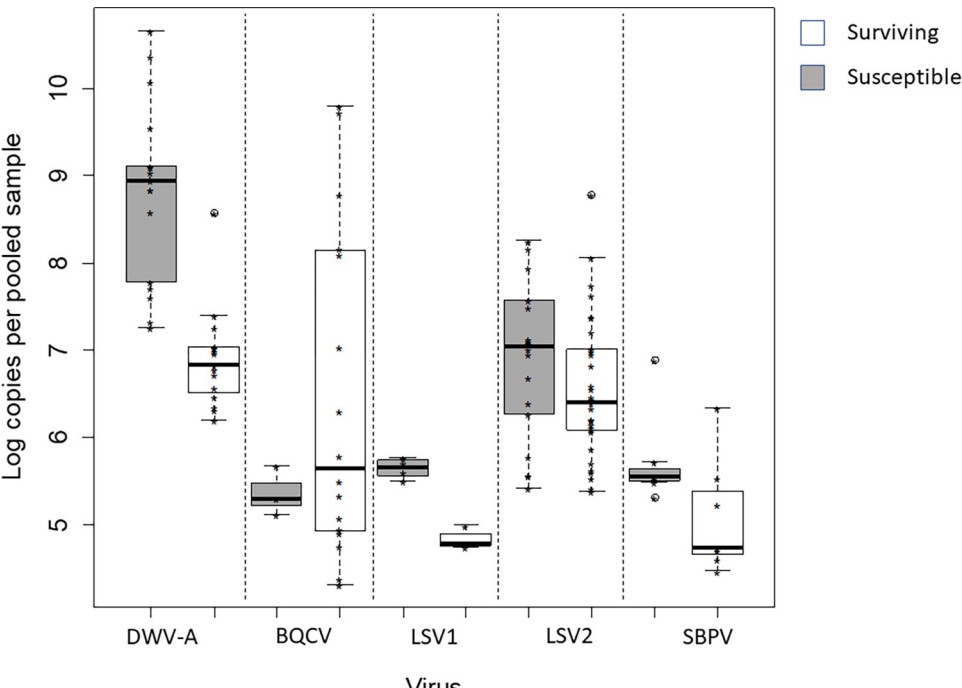

**Fig 3. Virus abundances in pooled adult worker samples of summer 2014 from surviving and susceptible *Apis mellifera* host colonies.** Medians, interquartile ranges, and maxima are shown. While there were no significant differences for most viruses (DWV-A, LSV2, SBPV) surviving colonies had significantly higher BQCV loads, compared to susceptible colonies (F = 39.64, df = 1, p < 0.001). Sampled colonies n = 19 susceptible and n = 20 surviving.

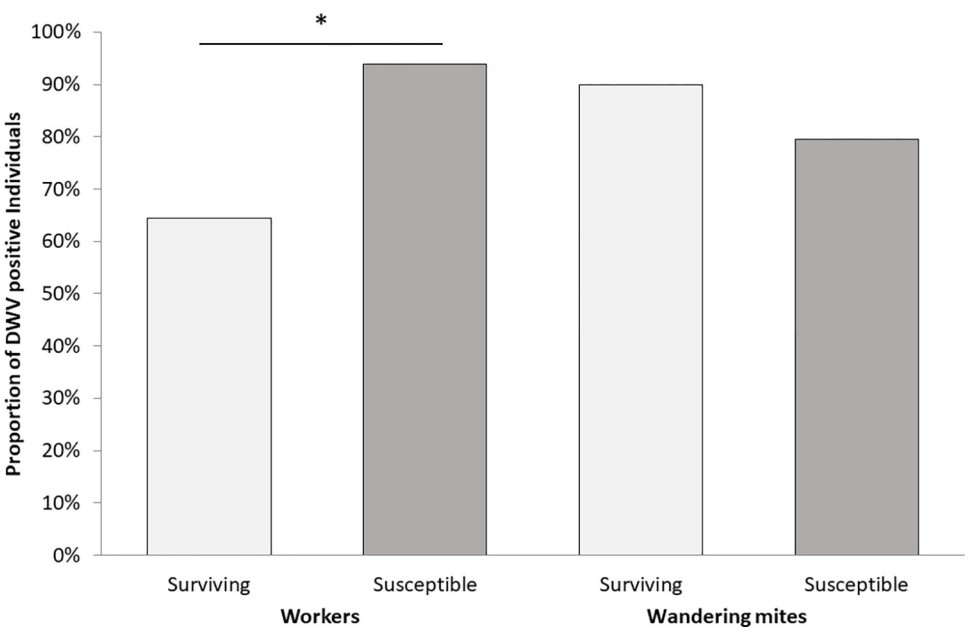

**Fig 4. Proportion of DWV-A-positive workers and phoretic mites collected from mite washes in spring, summer, and autumn.** Fewer adult worker bees in surviving colonies tested positive for DWV-A ($\chi^2$ = 33.751, df = 1, p < 0.01). Sampled colonies n = 29 surviving and n = 18 susceptible.

**Individual pupae and associated mite samples.** Viruses detected in brood and associated mites were DWV-A, DWV-B and SBPV, all confirmed using PCR sequencing. BQCV, LSV1 and LSV2 were not tested in individual pupae and mite samples.

*Virus prevalence.* Pupae sampled in surviving colonies had a lower prevalence of DWV-A ($\chi^2$ = 6.75, df = 1, p = 0.009, Table in S1 Table) than susceptible colonies. Population type (surviving or susceptible) did not significantly influence the prevalence of any other tested virus (Table in S1 Table). DWV-A and SBPV varied by season (DWV-A: $\chi^2$ = 28.64, df = 2, p < 0.001, SBPV: $\chi^2$ = 16.44, df = 2, p < 0.001, Table in S1 Table) and DWV-B was affected by the age of the pupae (13.41, df = 1, p < 0.001, Table in S1 Table).

*Virus abundance.* DWV-A abundance was significantly lower in surviving pupae and mites (pupae: F = 19.78, df = 1, p < 0.001, mites: F = 11.13, df = 1, p = 0.001, Fig 5, Table in S1 Table). Abundances were highest in autumn of 2013 (F = 12.57, df = 2, p < 0.001, Table in S1 Table) and abundance varied with the age of the pupae (F = 13.2, df = 1, p < 0.001, Table in S1 Table). DWV-B and SBPV abundances did not differ significantly between population type (surviving or susceptible, Table in S1 Table).

## Discussion

The data show that the mite-vectored DWV-A is lower in both abundance and prevalence in the mite-surviving honeybee population in tandem with lower phoretic mite infestations compared to the mite-susceptible population, especially at the critical period in autumn. When adult worker bees were pooled, DWV-A prevalence was lower, but not abundance, meaning that when a bee was subjected to the virus, levels of the virus were similar to the controls. The data therefore suggest that general adaptations to virus infections, though still possible, are unlikely to explain colony survival. Instead, mechanisms suppressing mite reproduction and therefore reducing vector presence seem to be more important, a hypothesis put forward by Grindrod and Martin (2021) [32].

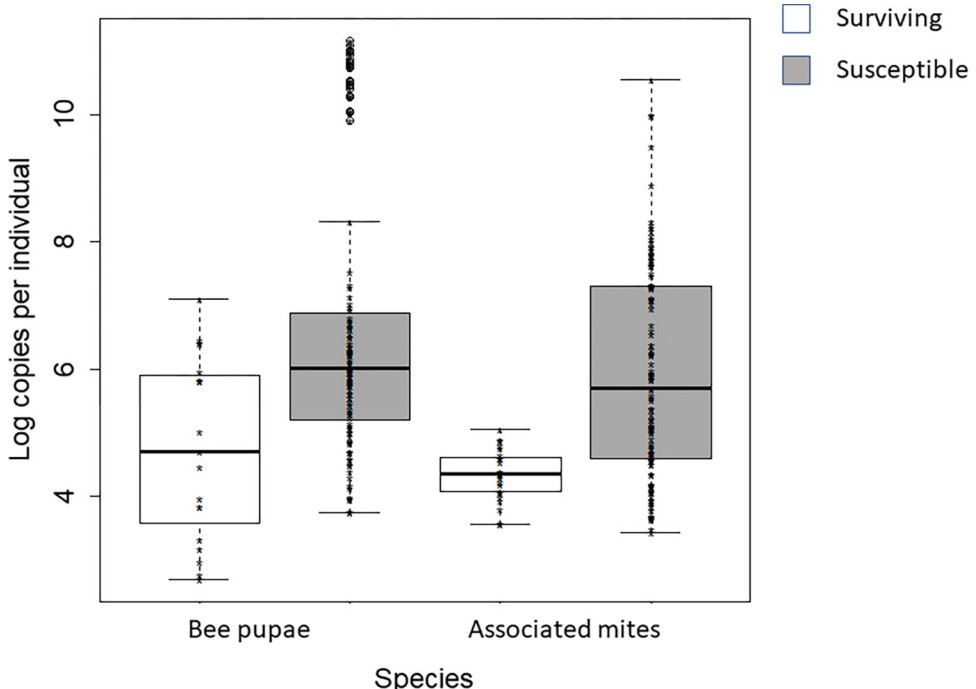

**Fig 5. Abundance of DWV-A in worker pupae and associated mites in autumn 2013 from surviving (untreated) and susceptible (treated) colonies.** Medians, interquartile ranges, and maxima are shown. While there was no significant difference in the proportions of DWV-A positive mites ($\chi^2$ = 2.455, df = 1, p > 0.05, sampled colonies n = 15 surviving and n = 19 susceptible), significantly fewer worker bees from surviving colonies had detectable DWV-A abundances compared to susceptible colonies ($\chi^2$ 33.751, df = 1, p < 0.01) Sampled colonies n = 29 surviving and n = 18 susceptible.

When compared to the sympatric, susceptible colonies, phoretic mite levels and DWV-A prevalence were lower in all collected samples. DWV-A abundance was lower in individual pupae and adult worker samples but showed no significant difference in pooled adult worker samples, meaning when an adult bee was bitten by a phoretic mite, the levels of virus may often have been the same as in mite-susceptible bees. In line with this: DWV-A abundance was also lower in mites collected from surviving colony brood but not lower in mites collected off adult bees, possibly pointing to horizontal transfer, and/or the vector's efficiency of transmission when given the chance to interact with the virus. DWV-A abundance in pooled worker samples did increase with the level of infested brood, pointing again to levels of the virus being dictated primarily by the presence of the vector. These findings fall in-line with the findings of previous studies, where viral loads in colonies displaying resistances to *V. destructor* were significantly lower than their susceptible counterparts, both in tested South African populations [33] and in Uruguayan populations [34].

The general pattern for mite populations in non-adapted Western honeybee colonies is a steady and exponential increase in numbers from spring to autumn [35]. Mite levels in this study were low in summer in susceptible colonies, and this may have been due to the subsequent replacement of dead colonies with new nucleus colonies when they became available in mid-June. For this reason, and likely because mite levels did not change significantly over season in surviving colonies, season was not a significant factor in dictating colony levels of *V. destructor*. Mite numbers measured in bee brood were not found to be different between population types; however, no brood samples were collected in autumn, the most critical period for measuring mite population loads. Previous research on the surviving population has found

that mite populations are lower in autumn than in sympatric control colonies regularly treated for mites [8].

The prevalence of DWV-A in pooled phoretic mite samples was just as high in the surviving bee population as it was in the susceptible population, even though prevalence was lower in bees, i.e. mites tested positive for DWV-A in comparative frequencies between populations, while the bees in surviving colonies tested positive less frequently. This evidence was reinforced in the individual adult bee samples and phoretic mite samples. This could suggest a viral resistance in bees; however, because pooled worker samples had comparable levels of DWV-A in both populations, this seems unlikely: some workers clearly contracted comparably high abundances of the virus in surviving colonies. Observed rates in adult worker pooled samples may be an indicator of horizontal mite spread from colonies outside of the experimental apiaries (comparable abundance) and the ability of bees to reduce the number of mites and therefore the probability of being bitten by foreign mites and contracting high loads of DWV-A (lower prevalence). Horizontal transmission is an element of the system that has not received nearly enough attention in the past and requires further investigation.

In mites, DWV-A abundance was significantly lower in the surviving population when they were taken from brood cells. This is very likely a result of reduced vector infestation levels in the surviving colonies and the consequent reduction of pathogen spread.

BQCV was only tested in the pooled adult worker samples, but both prevalence and abundance were significantly higher in surviving colonies. This data combined with comparative DWV-A abundances in pooled worker bee samples suggest that an adaptation centred around a general increase in response to virus infections may be unlikely to explain colony survival. The central mechanism seems to be a control of the viral vector, *V. destructor*. However, traits fostering the ability to survive mite infestations may render colonies more susceptible to other threats: The cell recapping behaviour seems to be involved in suppressing mite reproduction [21] and is present in all tested mite-surviving bee populations [9, 15, 36], but opening of brood cells may result in a higher susceptibility to other pathogens, such as BQCV.

DWV-B and SBPV abundances changed with season and DWV-B levels fluctuated with pupal age but were comparable between population types. This study was performed in 2014 and 2013, and it is likely DWV-B was not present in great levels in Norway at this time, though little prior data had been collected to confirm its presence or absence and was not published. Interestingly, LSV1 and LSV2 were mostly absent in mites however little can be said about the relationship of these viruses to the mite-surviving traits observed in the bees.

In conclusion, this study provides evidence that prevalence and abundance of *V. destructor*-associated viruses can be lower in honeybee populations adapted to the parasite vector [8, 15, 21]. The less-frequent the vector, the lower the chance of viral transmission [32]. Viral tolerance cannot be completely excluded, as it is known that the ability to cope with *V. destructor* infestations is likely a product of several interacting traits [10, 37], and interactions between viruses merit more attention in literature. The higher levels of BQCV are in line with previous observations made within this population, though this is the first time quantifiable data of this pattern have been obtained. In another surviving population in Sweden, BQCV abundances were substantially lower compared to a local susceptible population [38]. This may point to different mechanisms enabling colony survival in the various surviving populations, and the independence of the selection pathways. This further suggests that adaptations to *V. destructor* may render colonies more vulnerable to other pathogens. Such a trade-off scenario seems likely, as higher levels of cell recapping in mite-surviving populations [8–15] increases exposure risk to the more vulnerable brood.

The presence of suppressed mite reproduction in all recorded mite-surviving Western honeybee populations so far [7, 8, 39] suggests that this is the most successful natural strategy

enabling survival of infested host colonies, though care must be taken to account for local challenges and it is always better to focus on the adaptive potential of regional stocks [40, 41]. Viral tolerance cannot be discounted however, and we must also consider the adaptive potential of mites, future studies on *V. destructor*-survivability in honeybees may benefit from a focus on the bees' ability to regulate the parasite and not endure the mites and associated viral infections.

## Supporting information

**S1 Table. Model output for the full analysis.** Parameters include: the population type (treated or untreated), "Season" is the time of year the samples were taken (Spring, Summer or Autumn), "Species", whether the organism screened was a honeybee or a V. destructor mite and "Brood/Worker infestation" The V. destructor infestation rate sampled per colony in either brood or adult worker bees.
(PDF)

**S2 Table. Primers used for the qualitative and quantitative detection of bee viruses.**
(PDF)

**S1 File. Virus infections in honeybee colonies naturally surviving ectoparasitic mite vectors.**
(DOCX)

**S1 Data.**
(XLSX)

**S2 Data.**
(XLSX)

**S3 Data.**
(XLSX)

**S4 Data.**
(XLSX)

**S5 Data.**
(XLSX)

**S6 Data.**
(XLSX)

## Acknowledgments

We are grateful to the cooperating beekeepers that allowed us to collect samples from their honeybee colonies.

## Author Contributions

**Conceptualization:** Melissa A. Y. Oddie, Sandra Lanz, Peter Neumann.

**Data curation:** Melissa A. Y. Oddie, Sandra Lanz, Orlando Yañez.

**Formal analysis:** Melissa A. Y. Oddie, Sandra Lanz.

**Funding acquisition:** Bjørn Dahle, Peter Neumann.

**Investigation:** Melissa A. Y. Oddie.

**Methodology:** Melissa A. Y. Oddie, Orlando Yañez.

**Project administration:** Bjørn Dahle, Peter Neumann.

**Supervision:** Bjørn Dahle, Peter Neumann.

**Validation:** Orlando Yañez.

**Visualization:** Melissa A. Y. Oddie.

**Writing – original draft:** Melissa A. Y. Oddie.

**Writing – review & editing:** Melissa A. Y. Oddie, Bjørn Dahle, Peter Neumann.

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
