## [Decision Letter · Decision Letter 0]

18 Jan 2023

PONE-D-22-33692Virus infections in honeybee colonies naturally surviving ectoparasitic mite vectorsPLOS ONE

Dear Dr. Oddie,

Thank you for submitting your manuscript to PLOS ONE. After careful consideration, we feel that it has merit but does not meet PLOS ONE’s publication criteria as it currently stands. Therefore, we invite you to submit a revised version of the manuscript that addresses the points raised during the review process. Specifically, reviewer 2 was very critical and has explained several important issues. They are concerned about the rigor of your methods and terminology and also raised the possibility that this manuscript was previously reviewed and then submitted here without sufficiently addressing previous reviews. This would be a waste of everybody's time and therefore I would urge you to careful pay attention to the criticisms of that reviewer if you chose to revise and resubmit.

We look forward to receiving your revised manuscript.

Kind regards,

Olav Rueppell

Academic Editor

PLOS ONE

Journal Requirements:

Reviewers' comments:

Reviewer's Responses to Questions

**Comments to the Author**

1. Is the manuscript technically sound, and do the data support the conclusions?

Reviewer #1: Yes

Reviewer #2: No

2. Has the statistical analysis been performed appropriately and rigorously? 

Reviewer #1: Yes

Reviewer #2: I Don't Know

3. Have the authors made all data underlying the findings in their manuscript fully available?

Reviewer #1: Yes

Reviewer #2: Yes

4. Is the manuscript presented in an intelligible fashion and written in standard English?

Reviewer #1: Yes

Reviewer #2: Yes

5. Review Comments to the Author

Reviewer #1: The paper by Oddie et al. focuses on the analysis of viral infections (prevalence and viral loads) in Norwegian honey bee colonies that naturally survived infestation by Varroa destructor. They found lower V. destructor infestation levels in surviving colonies compared to Varroa susceptible colonies, and detected BQCV, DWV-A, LSV1, LSV2 and SBPV in all the colonies. The prevalence and viral load of Deformed Wing Virus (DWV-A) was lower in the mite surviving colonies in contrast to BQCV. Since the former virus is vectored by the mite they concluded that their findings suggests that lower mite levels may account for reduced prevalence and titers DWV-A rather than other mechanism of virus resistance in the surviving colonies

The study is interesting, well performed and described and it contributes to our knowledge on virus-resistance observed in V. destructor naturally surviving honey bee colonies. The conclusion is sound and it paves the way for further investigation of these mechanisms at the colony and individual bee levels.

The statement at lines 79-81 that indicates that the IKA viruses (IAPV, KBV and ABPV) and SBPV are not vectored by V. destructor should be changed since these viruses can be vectored by the mite as stated in Yañez et al. 2020, Bee Viruses: Routes of Infection in Hymenoptera.

Reviewer #2: Editors,

The manuscript entitled, “Virus infections in honeybee colonies naturally surviving ectoparasitic mite vectors” by Oddie et. al describes differences in virus abundance between a reported samples obtained from colonies that are reportedly mite-resistant (i.e., survivor honey bee stock) compared to samples obtained from conventionally managed colonies. This is an interesting topic, but as currently presented it was too difficult to interpret the data to ensure accuracy, and the text requires additional work prior to publication.

Points to clarify or address before publication include:

1. Figures – In general, virus abundance data be better presented as box and whisker graphs that include and data point for all samples (i.e. rather than bar charts), so that readers can get a better understanding of the values and actual variability in the data.

2. Abstract Line 21 / Introduction – While the citations in the introduction provide insight into mechanisms / behaviors associated with mite resistance and indicate that these adaptations can occur within 5 years (which seems short, if selection is acting at the colony level). It would be good for the authors to elaborate on this topic in regard to selection at the individual and/or colony level, etc. and/or revise the first line in their abstract and other place in the text. The researchers should be in other examples of hosts that have developed resistant to parasites over a similar relative time period (i.e., discuss typically ‘generation time’ for a honey bee colony, which is likely 1-2 years+ , and the number of years of mite exposure in Norway, etc.) in order to better describe how the mite-resistance described in this manuscript is/may be the result of natural selection.

3. Abstract Line 22 - The Abstract should be changed since if the colonies were selected from mite resistance, that may result in lower viral load since mites are amplifying vectors of several honey bee viruses and likely passive vectors of many others – but that does not mean that the honey bees have developed resistance to the viruses. While the Abstract gets to this point it begins by stating “It is possible that resistance to the viruses spread by this parasite may contribute to colony survival,. . .” which is misleading/ confusing.

4. Abstract – “The damaging variant of DWV, (DWV-A) .. “ is incorrect, both DWV-A and DWV-B can be detrimental to honey bees, as well as recombinant viruses.

5. The authors need to include a list of primers used in this study, including the primers used to distinguish DWV-A vs. DWV-B. Were the viruses sequence? Were DWV-recombinants assessed? Without this information, some of the DWV abundance in this study may not be accounted for, and therefore the conclusions would only be based on some of the DWV. Regional differences in RNA viruses, which have required new primers for detection and quantification may be needed (e.g., see Daughenbaugh et al Viruses 2021 doi: 10.3390/v13020291, Moore et al. J. Gen. Virol. 2010 doi: 10.1099/vir.0.025965-0, and others).

6. The Virus nomenclature needs to be revised throughout the manuscript, Lines 77-80+, to match ICTV guidelines. Virus nomenclature suggestion based on ICTV: , “black queen cell virus” and “deformed wing virus”, etc. be written in lower case, when referred to in-general (i.e., not a specific strain or species) should be written in lower case and not-italicized.

See: https://talk.ictvonline.org/information/w/faq/386/how-to-write-virus-species-and-other-taxa-names

Considering “deformed wing virus-A” is a virus name, “A virus name should never be italicized, even when it includes the name of a host species or genus, and should be written in lower case. The first letters of words in a virus name, including the first word, should only begin with a capital when these words are proper nouns (including host genus names but not virus genus names) or start a sentence.” Alternatively, for specific virus species, the authors could use “Deformed wing virus-A” (i.e., first letter capitalized, and italicized word), but this is rarely used.

7. Figures, were all colonies assessed on the same date? The supplemental information is good, but would be improved by inclusion of sample date (not just season) or at least including the exact sample dates in the methods. Since the virus abundance in any colony changes over time.

8. The use of the word “titres” is inaccurate and should be revised throughout the manuscript.

Virus titer or viral titre = titre typically refers to the concentration of infectious viral particles

The authors present “viral abundance” or “viral load” data in the form of RNA or cDNA copies per bee (which would be better presented as copies/ xxx ng RNA), therefore the text should be revised.

9. Virus abundance data is a major component of this study – yet it is unclear how this was determined in this study, and the reporting on figures, etc. requires major modification.

Specifically –

(a) Figure 3 – Log copies per colony is not the correct axis label.

It should be log(10) copies per 1000 ng total RNA (if they scaled results to represent total RT reaction).

The reaction volumes for RT and qPCR reactions should be included in the methods section and it should be clearly stated if Line 139 “3 ul cDNA” is really “3 ul of the 1:10 cDNA” – since, as written it is impossible to compare the virus copy number reported in this paper to other studies.

The researchers do not know the copy number per colony, since the assessment is done per x ul of a cDNA reaction – it represents some amount of RNA sample obtained from a subset of bees sampled from one colony.

(b) Lines 146 – 152, it is unclear how virus standards were made since simple dilutions of an unknown stock would not work to calculate “virus copy numbers” .

What did the authors use as a virus standard?

The standard data should be included and graphed to evaluate copy number and the qPCR efficiency.

PCR efficiency typically expressed as a percentage

(i.e., 10^(1/slope “3.32) – 1 = 1 or 100%

I am not sure how the authors calculated efficiency.

(c) The qPCR standard curves used to calculate virus abundance per qPCR well, and the math required to make that number representative of a given amount or RNA need to be clearly presented in this manuscript.

10. Virus abundance in individual colonies varies over time within a colony and the variation between individual colonies with the same apiary and/or region can be high.

The sample size in this study, which was n=10 mite-tolerant colonies, and n=11 conventionally managed colonies for virus abundance data was low. This sample sizes should clearly stated in the figures / figure legends The methods section reports higher numbers of both colonies in other sections making it difficult to see that the virus data came from a subset of the total number of colonies described.

The authors should revise the text describing putative virus resistance vs. tolerance since with a total of 21 colonies (i.e., 10 or 11 colonies per group) it would be difficult to try to assess the potential of virus resistance vs. tolerance.

For pupae and mite assessment it seems, 29 samples were obtained from mite resistant colonies located in three apiaries and 18 conventionally managed mite-susceptible colonies from five apiaries.

Since sample sizes vary for each figure, they must be included in the figures and figure legends, they should be included in every figure or figure legend.

11. Discussion – Although the virus levels reported in honey bees in this study were lower in the “mite-resistant stock” were lower than the “conventionally managed susceptible stock” the number of mites were also lower – so the conclusion stated at the beginning of the discussion may not be accurate (i.e., Line 252), lower mite numbers may be associated with lower virus numbers.

12. As written the discussion is difficult to read, and included big overarching conclusions based on the data obtained from a relatively few number of colonies – that may or may not be exhibiting “mite-resistance” phenotype given that the levels of mite in the brood are similar in the different stocks (although they are missing the autumn time point).

There may be other areas to comment on in the discussion, but since the virus data was difficult to review/interpret it is not possible to fully evaluate the discussion.

Minor points to clarify or address before publication include:

1. Fig. 2 – y-axis does not need decimal places (e.g., 10%, instead of 10.00%)

6. PLOS authors have the option to publish the peer review history of their article (what does this mean?). If published, this will include your full peer review and any attached files.

Reviewer #1: **Yes: **Nor Chejanovsky

Reviewer #2: No

---

## [Author Response · Author response to Decision Letter 0]

24 Mar 2023

Reviewer #1: The paper by Oddie et al. focuses on the analysis of viral infections (prevalence and viral loads) in Norwegian honey bee colonies that naturally survived infestation by Varroa destructor. They found lower V. destructor infestation levels in surviving colonies compared to Varroa susceptible colonies, and detected BQCV, DWV-A, LSV1, LSV2 and SBPV in all the colonies. The prevalence and viral load of Deformed Wing Virus (DWV-A) was lower in the mite surviving colonies in contrast to BQCV. Since the former virus is vectored by the mite they concluded that their findings suggests that lower mite levels may account for reduced prevalence and titers DWV-A rather than other mechanism of virus resistance in the surviving colonies

The study is interesting, well performed and described and it contributes to our knowledge on virus-resistance observed in V. destructor naturally surviving honey bee colonies. The conclusion is sound and it paves the way for further investigation of these mechanisms at the colony and individual bee levels.

The statement at lines 79-81 that indicates that the IKA viruses (IAPV, KBV and ABPV) and SBPV are not vectored by V. destructor should be changed since these viruses can be vectored by the mite as stated in Yañez et al. 2020, Bee Viruses: Routes of Infection in Hymenoptera.

Response: We firstly thank the reviewer very much for their time and effort. This correction has been made and the paper added to citations.

Reviewer #2: Editors,

The manuscript entitled, “Virus infections in honeybee colonies naturally surviving ectoparasitic mite vectors” by Oddie et. al describes differences in virus abundance between a reported samples obtained from colonies that are reportedly mite-resistant (i.e., survivor honey bee stock) compared to samples obtained from conventionally managed colonies. This is an interesting topic, but as currently presented it was too difficult to interpret the data to ensure accuracy, and the text requires additional work prior to publication.

Points to clarify or address before publication include:

1. Figures – In general, virus abundance data be better presented as box and whisker graphs that include and data point for all samples (i.e. rather than bar charts), so that readers can get a better understanding of the values and actual variability in the data.

Response: We thank the reviewer for thorough work and have endeavoured to address all points made:

We have added the points to the box and whisker plots as advised.

2. Abstract Line 21 / Introduction – While the citations in the introduction provide insight into mechanisms / behaviors associated with mite resistance and indicate that these adaptations can occur within 5 years (which seems short, if selection is acting at the colony level). It would be good for the authors to elaborate on this topic in regard to selection at the individual and/or colony level, etc. and/or revise the first line in their abstract and other place in the text. The researchers should be in other examples of hosts that have developed resistant to parasites over a similar relative time period (i.e., discuss typically ‘generation time’ for a honey bee colony, which is likely 1-2 years+ , and the number of years of mite exposure in Norway, etc.) in order to better describe how the mite-resistance described in this manuscript is/may be the result of natural selection.

Response: We thank the reviewer for highlighting these evolutionary aspects. There is consensus that European (-derived) populations of western honeybees, Apis mellifera, can survive infestations of Varroa destructor by means of natural selection This has been reviewed in detail by Locke 2016 and confirmed since, e.g. Oddie et al 2017 among many others. Clearly, selection at both individual bee and colony level are important in this regard as is the rather long generation time of honeybees. However, this has been dealt with in detail elsewhere (Locke 2016). So, here we just mention this briefly, cite the respective review accordingly, and prefer instead to focus on the possible role of the ability to cope with virus infections contributing to colony survival.

3. Abstract Line 22 - The Abstract should be changed since if the colonies were selected from mite resistance, that may result in lower viral load since mites are amplifying vectors of several honey bee viruses and likely passive vectors of many others – but that does not mean that the honey bees have developed resistance to the viruses. While the Abstract gets to this point it begins by stating “It is possible that resistance to the viruses spread by this parasite may contribute to colony survival,. . .” which is misleading/ confusing.

Response: We are afraid that the reviewer may have misinterpreted part of the matter: The surviving colonies investigated in this study were not selected for any mite resistance whatsoever. Instead, natural selection has enabled colony survival which may result from resistance / tolerance to mite infestations and our virus infections. 

However, the reviewer is correct in another point: our main hypothesis is that viruses which are vectored by this mite should correlate with vector infestations levels and those not vectored by this mite should not. If the ability to cope with virus infections is an important factor contributing to colony survival, then virus levels should in general be low regardless of vector abundance and independent of a virus being vectored or not. On the other hand given that virus levels positively correlate with vector abundance and are equally high or even higher in non-mite-vectored viruses compared to mite-vectored ones, than the ability to generally cope with virus infections seems to be less relevant for colony survival than the ability to supress mite reproduction. Our data do indicate the latter. We have now rewritten the abstract to make this clearer to the readership. 

The reviewer is also absolutely correct about the terminology. From our data, it cannot be concluded about resistance / tolerance whatsoever. Instead, we stick to "mite-surviving" as a well known colony phenotype regardless of the actual underlying mechanism. 

4. Abstract – “The damaging variant of DWV, (DWV-A) .. “ is incorrect, both DWV-A and DWV-B can be detrimental to honey bees, as well as recombinant viruses.

Response: We have changed the terminology here to reflect the reviewer’s comment.

5. The authors need to include a list of primers used in this study, including the primers used to distinguish DWV-A vs. DWV-B. Were the viruses sequence? Were DWV-recombinants assessed? Without this information, some of the DWV abundance in this study may not be accounted for, and therefore the conclusions would only be based on some of the DWV. Regional differences in RNA viruses, which have required new primers for detection and quantification may be needed (e.g., see Daughenbaugh et al Viruses 2021 doi: 10.3390/v13020291, Moore et al. J. Gen. Virol. 2010 doi: 10.1099/vir.0.025965-0, and others).

Response: The paper now has a primer table in the supplementary files. More detailed information about the virus sequences were added to the text.

6. The Virus nomenclature needs to be revised throughout the manuscript, Lines 77-80+, to match ICTV guidelines. Virus nomenclature suggestion based on ICTV: , “black queen cell virus” and “deformed wing virus”, etc. be written in lower case, when referred to in-general (i.e., not a specific strain or species) should be written in lower case and not-italicized.

See: https://talk.ictvonline.org/information/w/faq/386/how-to-write-virus-species-and-other-taxa-names

Considering “deformed wing virus-A” is a virus name, “A virus name should never be italicized, even when it includes the name of a host species or genus, and should be written in lower case. The first letters of words in a virus name, including the first word, should only begin with a capital when these words are proper nouns (including host genus names but not virus genus names) or start a sentence.” Alternatively, for specific virus species, the authors could use “Deformed wing virus-A” (i.e., first letter capitalized, and italicized word), but this is rarely used.

Response: The virus nomenclature has now been carefully considered throughout the entire MS. 

7. Figures, were all colonies assessed on the same date? The supplemental information is good, but would be improved by inclusion of sample date (not just season) or at least including the exact sample dates in the methods. Since the virus abundance in any colony changes over time.

Response: We thank the reviewer for highlighting this important point. Yes, indeed, seasonal fluctuations in virus levels are well-known. Therefore, we have of course taken this into account and sampled both mite-surviving and mite susceptible colonies in the same months. This has now been added.

8. The use of the word “titres” is inaccurate and should be revised throughout the manuscript.

Virus titer or viral titre = titre typically refers to the concentration of infectious viral particles

The authors present “viral abundance” or “viral load” data in the form of RNA or cDNA copies per bee (which would be better presented as copies/ xxx ng RNA), therefore the text should be revised.

Response: The entire text has been revised accordingly. 

9. Virus abundance data is a major component of this study – yet it is unclear how this was determined in this study, and the reporting on figures, etc. requires major modification.

Specifically –

(a) Figure 3 – Log copies per colony is not the correct axis label.

It should be log(10) copies per 1000 ng total RNA (if they scaled results to represent total RT reaction).

The reaction volumes for RT and qPCR reactions should be included in the methods section and it should be clearly stated if Line 139 “3 ul cDNA” is really “3 ul of the 1:10 cDNA” – since, as written it is impossible to compare the virus copy number reported in this paper to other studies.

The researchers do not know the copy number per colony, since the assessment is done per x ul of a cDNA reaction – it represents some amount of RNA sample obtained from a subset of bees sampled from one colony.

(b) Lines 146 – 152, it is unclear how virus standards were made since simple dilutions of an unknown stock would not work to calculate “virus copy numbers” .

What did the authors use as a virus standard?

The standard data should be included and graphed to evaluate copy number and the qPCR efficiency.

PCR efficiency typically expressed as a percentage

(i.e., 10^(1/slope “3.32) – 1 = 1 or 100%

I am not sure how the authors calculated efficiency.

(c) The qPCR standard curves used to calculate virus abundance per qPCR well, and the math required to make that number representative of a given amount or RNA need to be clearly presented in this manuscript.

Response: We have addressed all of these concerns accordingly. In particular, the axis label on figure 3 has been changed to “Log copies per pooled sample”. More detailed information about the qPCR efficiencies and the standard curves were added to the text.

10. Virus abundance in individual colonies varies over time within a colony and the variation between individual colonies with the same apiary and/or region can be high.

The sample size in this study, which was n=10 mite-tolerant colonies, and n=11 conventionally managed colonies for virus abundance data was low. This sample sizes should clearly stated in the figures / figure legends The methods section reports higher numbers of both colonies in other sections making it difficult to see that the virus data came from a subset of the total number of colonies described.

The authors should revise the text describing putative virus resistance vs. tolerance since with a total of 21 colonies (i.e., 10 or 11 colonies per group) it would be difficult to try to assess the potential of virus resistance vs. tolerance.

For pupae and mite assessment it seems, 29 samples were obtained from mite resistant colonies located in three apiaries and 18 conventionally managed mite-susceptible colonies from five apiaries.

Since sample sizes vary for each figure, they must be included in the figures and figure legends, they should be included in every figure or figure legend.

Response: We agree with the sample size issue and are now much more careful in our data interpretation. Nevertheless, we are convinced that our data are solid. Again, we also agree with the virus resistance vs. tolerance issue (see above) and have removed this from the entire text. Finally, sample sizes have been added in the figure caption for each figure.

11. Discussion – Although the virus levels reported in honey bees in this study were lower in the “mite-resistant stock” were lower than the “conventionally managed susceptible stock” the number of mites were also lower – so the conclusion stated at the beginning of the discussion may not be accurate (i.e., Line 252), lower mite numbers may be associated with lower virus numbers.

Response: Again, we fully agree. The wording has been changed. Yes indeed, in our data low mite numbers correlate with low levels of the mite-vectored DWV-A thereby nicely confirming our hypothesis. We have now reworded accordingly. 

12. As written the discussion is difficult to read, and included big overarching conclusions based on the data obtained from a relatively few number of colonies – that may or may not be exhibiting “mite-resistance” phenotype given that the levels of mite in the brood are similar in the different stocks (although they are missing the autumn time point).

There may be other areas to comment on in the discussion, but since the virus data was difficult to review/interpret it is not possible to fully evaluate the discussion.

Response: The mite survival of the colonies used for this experiment has been confirmed earlier and is referenced extensively in the introduction and discussion. In any case, we have substantially revised the discussion to make it clearer and more concise. In particular, we are much more careful now with our conclusions based on the sample size matter raised above. 

Minor points to clarify or address before publication include:

1. Fig. 2 – y-axis does not need decimal places (e.g., 10%, instead of 10.00%)

Response: This has been addressed.

---

## [Decision Letter · Decision Letter 1]

11 Apr 2023

PONE-D-22-33692R1Virus infections in honeybee colonies naturally surviving ectoparasitic mite vectorsPLOS ONE

Dear Dr. Oddie,

Thank you for submitting your manuscript to PLOS ONE. As you can see below, I sent the manuscript out to one of the previous reviewers who was very critical before but is now almost satisfied. Therefore, we invite you to submit a revised version of the manuscript that takes their remaining concerns into account. One issue that was not directly stated in the review but was brought to my attention was your assertion in line 45. While I (and the reviewer) agree that the potential exists, I would suggest to caution readers that this potential may not be realized. Encouraging "Darwinian beekeeping" may be your goal, but I think it is only fair to put in more cautious language for beekeepers that may well read an open-access article and gamble their livelihood on that potential. The survivor populations exist, but they are not necessarily the rule (which makes them particularly interesting).Also, please check for spelling and other minor mistakes throughout. Some mistakes are not caught by automatic spell-checkers (e.g., line 65: "be" instead of "me").

We look forward to receiving your revised manuscript.

Kind regards,

Olav Rueppell

Academic Editor

PLOS ONE

Journal Requirements:

Reviewers' comments:

Reviewer's Responses to Questions

**Comments to the Author**

1. If the authors have adequately addressed your comments raised in a previous round of review and you feel that this manuscript is now acceptable for publication, you may indicate that here to bypass the “Comments to the Author” section, enter your conflict of interest statement in the “Confidential to Editor” section, and submit your "Accept" recommendation.

Reviewer #2: All comments have been addressed

2. Is the manuscript technically sound, and do the data support the conclusions?

Reviewer #2: Partly

3. Has the statistical analysis been performed appropriately and rigorously? 

Reviewer #2: Yes

4. Have the authors made all data underlying the findings in their manuscript fully available?

Reviewer #2: Yes

5. Is the manuscript presented in an intelligible fashion and written in standard English?

Reviewer #2: (No Response)

6. Review Comments to the Author

Reviewer #2: 

The manuscript entitled, “Virus infections in honeybee colonies naturally surviving ectoparasitic mite vectors” by Oddie et. al was significantly improved by the review process.

Points to clarify or address before publication include:

1. While I understand that the abstract “European honeybee populations,

Apis mellifera, “ is indicating “Apis mellifera” populations in Europe, since some papers still define Apis mellifera as the European honey bee (as opposed to the Western honey bee) – the authors should revise the abstract to make sure this is clear – since PONE has a general readership.

7. PLOS authors have the option to publish the peer review history of their article (what does this mean?). If published, this will include your full peer review and any attached files.

Reviewer #2: No

---

## [Author Response · Author response to Decision Letter 1]

13 Apr 2023

Response to reviewer

Reviewer #2 comments:

The manuscript entitled, “Virus infections in honeybee colonies naturally surviving ectoparasitic mite vectors” by Oddie et. al was significantly improved by the review process.

Points to clarify or address before publication include:

1. While I understand that the abstract “European honeybee populations,

Apis mellifera, “ is indicating “Apis mellifera” populations in Europe, since some papers still define Apis mellifera as the European honey bee (as opposed to the Western honey bee) – the authors should revise the abstract to make sure this is clear – since PONE has a general readership.

Response:

The term in question meant to be very general and is meant to include all populations of Apis mellifera, as it is known that A. m. scutellata is very capable of persisting through V. destructor infestations. However, since this paper is wholly about a population actually bred in Europe, we have made this distinction.

---

## [Decision Letter · Decision Letter 2]

9 May 2023

PONE-D-22-33692R2

Virus infections in honeybee colonies naturally surviving ectoparasitic mite vectors

PLOS ONE

Dear Dr. Oddie,

Thank you for submitting your manuscript to PLOS ONE. After careful consideration, we feel that it has merit but does not fully meet PLOS ONE’s publication criteria as it currently stands. Therefore, we invite you to submit a revised version of the manuscript that addresses the points raised during the review process.

We look forward to receiving your revised manuscript.

Kind regards,

Olav Rueppell

Academic Editor

PLOS ONE

Journal Requirements:

Additional Editor Comments:

Thank you for re-submitting your manuscript to PLOS ONE. As indicated in our previous correspondence, I felt obligated to it out for additional review and I was fortunately able to quickly secure two additional reviewers. As you can see, both have additional comments and concerns. While I do not think that the demand of reviewer #3 for additional experiments is warranted, I do think that both reviewers raise many important points. Most importantly, they confirm my concern about the language that could mislead beekeepers and cause them to stop treating for Varroa. I quote an excerpt from their confidential comments to the editor in this case: "The most important point, however, is that the wording concerning the evolution of mite tolerance must be adjusted (first sentence of the abstract and l. 45-47 in the Introduction)." Reviewer #4 also indicates a number of other necessary changes (for example Table 1 now seems to be missing from the manuscript?). If you are prepared to perform the necessary revisions, please follow the directions below. If you are not willing to do so, please just indicate this in an email to save the work of another submission because I will be forced to reject the manuscript in that case.

Reviewers' comments:

Reviewer's Responses to Questions

**Comments to the Author**

1. If the authors have adequately addressed your comments raised in a previous round of review and you feel that this manuscript is now acceptable for publication, you may indicate that here to bypass the “Comments to the Author” section, enter your conflict of interest statement in the “Confidential to Editor” section, and submit your "Accept" recommendation.

Reviewer #3: (No Response)

Reviewer #4: (No Response)

2. Is the manuscript technically sound, and do the data support the conclusions?

Reviewer #3: Partly

Reviewer #4: Yes

3. Has the statistical analysis been performed appropriately and rigorously? 

Reviewer #3: Yes

Reviewer #4: Yes

4. Have the authors made all data underlying the findings in their manuscript fully available?

Reviewer #3: Yes

Reviewer #4: Yes

5. Is the manuscript presented in an intelligible fashion and written in standard English?

Reviewer #3: Yes

Reviewer #4: Yes

6. Review Comments to the Author

Reviewer #3: The paper by Oddie et al. analyzes viral infections (prevalence and viral loads) in Norwegian honey bee colonies that apparently survived infestation by Varroa destructor naturally. However, as a reviewer, I find the provided information insufficient to verify the authors' statement. The term "naturally survival" is not explicitly defined, and there is no information on the management system or procedures involved in the study. The lack of fundamental information, such as the size and strength of the colonies at the beginning of the experiment, feeding schedules, and honey collection, makes it difficult to evaluate the article's argument thoroughly.

The researchers found lower V. destructor infestation levels in surviving colonies compared to Varroa-susceptible colonies, which confirms the results from Dr. Oddie's previous work. The researchers detected BQCV, DWV-A, LSV1, LSV2, and SBPV in all the colonies. The prevalence and viral load of Deformed Wing Virus (DWV-A) was lower in the mite-surviving colonies, in contrast to BQCV. The authors concluded that their findings suggest that reduced mite levels may account for the lower prevalence and titers of DWV-A, rather than other mechanisms of virus resistance in the surviving colonies.

The study is interesting, but caution should be exercised when interpreting the results. In honey bee research conducted in the field, it can be challenging to obtain reliable control groups. Thus, it is uncertain what diseases or conditions were initially present in the colonies at the beginning of the experiment, which could interfere with the intended measurements.

Although the authors measured some pathogens, it is clear that unmeasured pathogens interfere with the replication of another pathogen, making it difficult to draw firm conclusions.

My suggestion is to include the missing information management information and perhaps perform an additional experiment by injecting bees with purified viruses and measuring replication efficiency in these two honey bee populations.

This experiment could provide the authors with a more robust indication of resistance selection mechanisms than those proposed in this manuscript.

Without more information, any conclusion regarding the mechanisms by which these "naturally" surviving honey bees survive varroa mite infestations is pure conjecture.

Reviewer #4: In this study, the authors hypothesized that in honey bee populations that survive V. destructor infestations without treatment, that is, that are "mite survivors," the bees may also have increased tolerance or resistance to viral infections vectored by the mite. The results presented here from analyzing honey bee colonies originating from the "mite-surviving" population and from a "normal", hence, mite susceptible population over three successive seasons in 2013/2014 are interesting, the paper is nicely written, however, some amendments and corrections are still necessary:

l. 16-17: The wording needs to be amended. This reads as if Western honey bees in general are known to survive V. destructor infestation. But this is not the case. It is true that there are reports and research on a few populations that survive mite infestation without treatment, e.g. in France, Sweden, Norway. But - to the best of my knowledge - these "adapted" bees lost their "adaptation" once transferred to other environments/regions of Europe.

l. 19-20: This is a very simplistic view on virus infections and host tolerance. Host induced reduction of viral titers may be one possibility but sustaining higher viral titers may be another one and selection of less virulent virus variants may be yet another one.

Please reword the sentence and make clear that you only test one of several different possibilities here.

l. 25-26: What about DWV-B? That is the relevant, clearly mite associated DWV-variant. Why are you concentrating on the less virulent variant DWV-A? Please explain and justify your choice of variant.

l. 37: Please replace the term "recombinants" by the term "variants", because recombinants are only a subclass of the different variants within the DWV quasispecies.

l. 38: The wording is wrong: It is NOT the virus that developed new transmission routes but a new transmission route for DWV has opened up through the mite.

l. 45-47: The wording is misleading: Actually, small adapted populations have been reported from several European regions and have been analyzed for many years by now to identify the mechanisms behind these so far rather rare evolutionary events. Although these populations are worth to be studied, the sentence, as it reads right now, implies that every honey bee population would become "adapted" if left untreated. However, that such a development (evolution) would automatically occur within 5 years, if the colonies would be left untreated, has never been shown. Moreover, bee colonies from these allegedly mite-tolerant populations were no longer able to survive mite infestation without treatment when brought to other regions in Europe. The question still remains whether it is the bee or the mite or both that adapt to "lack of treatment".

Therefore, I request that the authors rephrase this sentence in a way that it better reflects the current knowledge and does not leave the impression that stopping mite treatment for five years will solve the mite problem and result in mite surviving honey bees all over Europe.

l. 54-55: Especially the Primorsky bees and the Gotland bees were tested for survival without Varroa treatment in Germany and they were no longer able to survive mite infestation without treatment. This has been published and should not be ignored because it is a very interesting aspect regarding the evolution of mite tolerance in honey bee populations.

l. 78: SBV and CBPV are known to be vectored by V. destructor, please read the literature.

That the vector capacity of the mite has not yet been shown for the other viruses mentioned does not mean that the mite is not able to vector them. There are just too few comprehensive studies on viruses in mites. Please correct and rephrase.

l. 172: It would be better to use the same terminology in the text and in the table, i.e. either worker infestation rate or phoretic mite infestation levels. Please correct.

l. 174: Will the supporting information (raw data in VirusVarroa.zip) also be available to the reader? If so, please refer here to this information but remove the names of the beekeepers in the file.

l. 182: (i) Please indicate whether mean +/- SD or +/- SEM is shown. (ii) The number of the colonies differs considerably between spring, summer, autumn. Why? Did colonies die? Did all "surviving" colonies really survive the winter season and how many of the "susceptible" colonies did not survive the winter season? (iii) Acc to MatMeth, the samples were collected in Sept 2013, April 2014 and June 2014. I assume that spring is April 2014, summer is June 2014 and autumn is Sept 2013. The proper order of the seasons in the figure then were autumn, spring, summer. Please rearrange and add the year. It will then become clear that information on winter mortality must exist because the colonies were observed over winter. Please provide this information. It is absolutely necessary for interpreting the data. Are the “surviving” colonies really surviving, are the susceptible colonies really more susceptible (more of them collapsing over winter) than the “surviving” colonies? Plesase add the requested information and consider it in the interpretation of your data.

l. 194: I assume that all colonies that were sampled in April and June 2014 survived the winter 2013/2014. Hence, strictly speaking, all colonies were "surviving" colonies". If this is not the case, please specify and differentiate between truely surviving colonies and replaced colonies in both groups.

I think it is better to keep writing mite-surviving and mite-susceptible colonies because otherwise the impression is given that the other colonies have not survived - but they had, because they were sampled!? Or didn't they? Then please specify.

Please amend the terminology throughout the manuscript.

l. 202, test stat (X2) 15.30: In Table 1, the value 15.03 is given. Please correct. Please explain "(from qpcr negative results)" given in the Table.

l. 203, test stat (X2) 6.75: In Table 1, the value 4.61 is given. Please correct.

l. 203, p value 0.009: In Table 1, the value 0,03 is given. Please correct.

l. 204: Acc to MatMeth, for virus prevalence pooled samples were collected in April and June 2014. Please provide this information somewhere in the legend of the Fig and explain/justify why you combined the results from two sampling time points.

l. 209, figure 4: It must be Fig. 3. Please correct.

l. 217: Please stick to one terminology! For instance, either write “mite-surviving colonies and mite-susceptible colonies” or “colonies from the mite-surviving population and the mite-susceptible population”, but don't switch between susceptible, mite-susceptible, treated etc. colonies.

This is especially relevant if you use a term here (treated) that cannot be found in the Figure or elsewhere.

l. 223: I cannot find results for individual adult workers in Table 1, only for (i) individual worker brood, (ii) pooled adult worker, and (iii) individual viral loads pupae: Likewise searching for X2 33.751 did not yield a result. Please correct.

l. 225: Please be consistent: If the legend of the axis says "wandering mites" don't use "phoretic mites" in the figure legend.

l. 238: For pupae, I can only find data for individual viral loads but not for prevalence in Table 1. Please correct.

l. 240: In Table 1 the term "viral load" is used, here it is "virus abundance". Please be consistent and rather take viral load throughout.

l. 246: Fig. 5 is about prevalence and abundance, not only prevalence. “Abundance” is better named viral load. Please correct.

l. 246: Fig. 5 is supposed to show surviving pupae and associated mites (see l. 241), not adult workers and phoretic mites. Please correct.

l. 247: Fig 5 is a box plot showing more than just means. Please describe correctly what is shown and specify the box and error bars (SD? SEM?).

l. 249: Here the use of "surviving" colonies is especially problematic. Since these data are from autumn, the use of the term surviving implies that the data of surviving colonies from both groups (mite survivjng and mite susceptible) are shown here, but I assume that surviving actually means “mite surviving”. Please correct.

l. 256: Please provide data on what happened over winter. Did the described differences had any influence on the mortality rate? Did the highly infested and virus infected colonies all die or did they survive? This information is lacking but highly relevant. Please add this information.

l. 259: Please also mention and properly discuss the possibility that the experimental design (pools of entire bees were analyzed, only DWV-A was analyzed instead of DWV-B or even better both variants) was not optimal for answering the question.

7. PLOS authors have the option to publish the peer review history of their article (what does this mean?). If published, this will include your full peer review and any attached files.

Reviewer #3: No

Reviewer #4: No

---

## [Author Response · Author response to Decision Letter 2]

20 Jun 2023

Reviewer #3: The paper by Oddie et al. analyzes viral infections (prevalence and viral loads) in Norwegian honey bee colonies that apparently survived infestation by Varroa destructor naturally. However, as a reviewer, I find the provided information insufficient to verify the authors' statement. The term "naturally survival" is not explicitly defined, and there is no information on the management system or procedures involved in the study. The lack of fundamental information, such as the size and strength of the colonies at the beginning of the experiment, feeding schedules, and honey collection, makes it difficult to evaluate the article's argument thoroughly.

The researchers found lower V. destructor infestation levels in surviving colonies compared to Varroa-susceptible colonies, which confirms the results from Dr. Oddie's previous work. The researchers detected BQCV, DWV-A, LSV1, LSV2, and SBPV in all the colonies. The prevalence and viral load of Deformed Wing Virus (DWV-A) was lower in the mite-surviving colonies, in contrast to BQCV. The authors concluded that their findings suggest that reduced mite levels may account for the lower prevalence and titers of DWV-A, rather than other mechanisms of virus resistance in the surviving colonies.

The study is interesting, but caution should be exercised when interpreting the results. In honey bee research conducted in the field, it can be challenging to obtain reliable control groups. Thus, it is uncertain what diseases or conditions were initially present in the colonies at the beginning of the experiment, which could interfere with the intended measurements.

Although the authors measured some pathogens, it is clear that unmeasured pathogens interfere with the replication of another pathogen, making it difficult to draw firm conclusions.

My suggestion is to include the missing information management information and perhaps perform an additional experiment by injecting bees with purified viruses and measuring replication efficiency in these two honey bee populations.

This experiment could provide the authors with a more robust indication of resistance selection mechanisms than those proposed in this manuscript.

Without more information, any conclusion regarding the mechanisms by which these "naturally" surviving honey bees survive varroa mite infestations is pure conjecture.

Response: 

We thank the reviewer for the time and effort they placed in reviewing our paper.

Thew reviewer’s argument is well-founded, viral loads shift over seasons and time. We have softened the language around the findings and removed the hypothesis on viral tolerance focusing instead on the expected pattern in viral loads for parasite suppression. 

Our observational studies were performed over the course of an entire year, so we do believe the most valuable shifts were captured in the data. We have provided further clarification as to the management and origins of the colonies, however the timing of management is defined by area, as temperature is dictated very much by the geography of the region and can separate seasonal patterns by as much as two weeks in as little a distance as 50km. The difficulty of standardizing management to collect meaningful data is an ever-present problem in field observations and should not be a deterrent to presenting observed patterns.

There are undoubtedly other pathogens affecting this system, however, the same can be said for all virus studies published, it is impossible to account for all interactions in a single study, and we feel this does not subtract from the legitimacy of the observational information presented in this paper.

The ability of this population to mitigate Varroa infestation levels is well-described in sourced papers and is not a subject of debate in this study.

We acknowledge that a manipulative experiment would support the findings but is not necessary in this case, as this is an observational study examining viral loads at the population level between two stocks with the fundamental difference of being adapted to V. destructor or not.

Reviewer #4: In this study, the authors hypothesized that in honey bee populations that survive V. destructor infestations without treatment, that is, that are "mite survivors," the bees may also have increased tolerance or resistance to viral infections vectored by the mite. The results presented here from analyzing honey bee colonies originating from the "mite-surviving" population and from a "normal", hence, mite susceptible population over three successive seasons in 2013/2014 are interesting, the paper is nicely written, however, some amendments and corrections are still necessary:

l. 16-17: The wording needs to be amended. This reads as if Western honey bees in general are known to survive V. destructor infestation. But this is not the case. It is true that there are reports and research on a few populations that survive mite infestation without treatment, e.g. in France, Sweden, Norway. But - to the best of my knowledge - these "adapted" bees lost their "adaptation" once transferred to other environments/regions of Europe.

Response: Firstly, we thank the reviewer for the time and effort they placed in reviewing our paper.

We have amended the abstract to clarify that we mean some populations and not all Western honeybees.

We are aware of a difference in experience between North American and European beekeeping in terms of Varroa destructor resistance, however the reviewer has received only a part of the picture presented by current literature: 

The claim that adapted bees “lose” their adaptation is untrue and unrealistic in the context of evolution. The changes in environment and Varroa density provide challenges that can reduce the effectiveness of traits that have been adapted for a specific environment, therefore it has always been recommended to focus on breeding resistance in locally-adapted bees. We have restated this now in the paper (line 330-332). 

There are many cases of untreated Western honeybees persisting in the face of Varroa, one of the latest published cases being the entire country of Cuba (Luis et al. 2022), where genetics studies revealed an almost exclusively European origin of the bees.

The argument of the loss of adaptation when bees are moved also falls drastically outside the scope of this paper, as there is no mention or encouragement of moving bees in any part of this paper.

l. 19-20: This is a very simplistic view on virus infections and host tolerance. Host induced reduction of viral titers may be one possibility but sustaining higher viral titers may be another one and selection of less virulent virus variants may be yet another one.

Please reword the sentence and make clear that you only test one of several different possibilities here.

Response: We have clarified that we mean to observe viral loads based on the known differences between these populations, we agree that this study is not sufficient to measure all possibilities of viral tolerance, we have also reworded the hypotheses to clarify our focus on the parasite suppression and its effect on viral loads.

l. 25-26: What about DWV-B? That is the relevant, clearly mite associated DWV-variant. Why are you concentrating on the less virulent variant DWV-A? Please explain and justify your choice of variant.

Response: DWV-B was discussed and tested for (line125), but when the data was collected it was not found in most of the samples. The reviewer must consider the spread of DWV-B and the timing of the study, coupled with the fact that Norway is a very closed country for honeybee imports and it is very possible this variant was not present in the population at the time.

l. 37: Please replace the term "recombinants" by the term "variants", because recombinants are only a subclass of the different variants within the DWV quasispecies.

Response: This has been amended.

l. 38: The wording is wrong: It is NOT the virus that developed new transmission routes but a new transmission route for DWV has opened up through the mite.

Response: This has been amended.

l. 45-47: The wording is misleading: Actually, small adapted populations have been reported from several European regions and have been analyzed for many years by now to identify the mechanisms behind these so far rather rare evolutionary events. Although these populations are worth to be studied, the sentence, as it reads right now, implies that every honey bee population would become "adapted" if left untreated. However, that such a development (evolution) would automatically occur within 5 years, if the colonies would be left untreated, has never been shown. Moreover, bee colonies from these allegedly mite-tolerant populations were no longer able to survive mite infestation without treatment when brought to other regions in Europe. The question still remains whether it is the bee or the mite or both that adapt to "lack of treatment".

Therefore, I request that the authors rephrase this sentence in a way that it better reflects the current knowledge and does not leave the impression that stopping mite treatment for five years will solve the mite problem and result in mite surviving honey bees all over Europe.

Response: We disagree with the reviewer’s claims here. There is a very abundant body of literature that provides evidence for the ability of Western honeybees to adapt to Varroa and we have cited it. We acknowledge the fundamental differences in experience between North American and European beekeeping operations, but this should not be a factor that dictates the interpretation of previous peer-reviewed evidence. Our wording was cautious, and “would develop” was not used, but “can” indicating the possibility but not a definite outcome. It has been observed to occur in as little as 5 years, this is in the literature cited. To the argument that honeybees “lose their adaptation when moved”, we refer to the argument and cited papers above.

We believe the language is sufficiently cautious given the evidence we have presented, there are no claims that stopping mite treatments will result in resistant bees indefinitely, and we will not be changing the wording here.

l. 54-55: Especially the Primorsky bees and the Gotland bees were tested for survival without Varroa treatment in Germany and they were no longer able to survive mite infestation without treatment. This has been published and should not be ignored because it is a very interesting aspect regarding the evolution of mite tolerance in honey bee populations.

Response: Here the reviewer likely refers to a paper written in 2004 by Berg et al. referenced below, however there was another study published on this same stock moved to the United States which showed a retention of the trait and the subsequent creation of the Russian Honeybee Breeder’s Association in the US (Rinderer et al. 2010), also referenced below.

We have cited both these papers and insist that Varroa resistance is a practical and realistic strategy as long as biology and local environments are considered.

Berg S., Fuchs S., Koeniger N., Rinderer T.E. (2004) Preliminary results on the comparison of Primorski honey bees, Apidologie 35, 552–554.

Rinderer TE, Harris JW, Hunt GJ, De Guzman LI: (2010) Breeding for resistance to Varroa destructor in North America. Apidologie 41:409-424 http://dx.doi.org/10.1051/apido/2010015.

We restate that the argument of moving bees does not apply to this paper, resistant bees have been shown to keep their resistance consistently when managed in the areas where they were bred. This has been documented in scientific literature in at least 8 independent cases across the globe.

l. 78: SBV and CBPV are known to be vectored by V. destructor, please read the literature.

That the vector capacity of the mite has not yet been shown for the other viruses mentioned does not mean that the mite is not able to vector them. There are just too few comprehensive studies on viruses in mites. Please correct and rephrase.

Response: The reviewer cannot claim that these viruses are vectored by Varroa, only associations have been shown in previous literature and these associations are not consistent. According to a paper published in 2022 (O’Shea-Wheller) CBPV can be associated with Varroa but viral transmission pathways have been “elusive”, citing two others: Seitz et al. (2019) and Ribière et al (2010), the same associations were found for SBV (Shen et al. 2005). We have edited the text to describe this association. If the reviewer insists on both viruses being actively vectored by Varroa, we ask that they provide literature where this has been shown, otherwise we must adhere to the consensus provided by the scientific literature found.

l. 172: It would be better to use the same terminology in the text and in the table, i.e. either worker infestation rate or phoretic mite infestation levels. Please correct.

Response: This has been amended.

l. 174: Will the supporting information (raw data in VirusVarroa.zip) also be available to the reader? If so, please refer here to this information but remove the names of the beekeepers in the file.

Response: All data is uploaded and available as per the journal’s requirements. The names of the beekeepers and locations have been changed to provide anonymity.

l. 182: (i) Please indicate whether mean +/- SD or +/- SEM is shown. (ii) The number of the colonies differs considerably between spring, summer, autumn. Why? Did colonies die? Did all "surviving" colonies really survive the winter season and how many of the "susceptible" colonies did not survive the winter season? (iii) Acc to MatMeth, the samples were collected in Sept 2013, April 2014 and June 2014. I assume that spring is April 2014, summer is June 2014 and autumn is Sept 2013. The proper order of the seasons in the figure then were autumn, spring, summer. Please rearrange and add the year. It will then become clear that information on winter mortality must exist because the colonies were observed over winter. Please provide this information. It is absolutely necessary for interpreting the data. Are the “surviving” colonies really surviving, are the susceptible colonies really more susceptible (more of them collapsing over winter) than the “surviving” colonies? Plesase add the requested information and consider it in the interpretation of your data.

Response: We have now indicated that the standard errors are shown.

The number of colonies differ due to reasons including loss, accessibility and beekeeper permissions, some colonies are moved onto heather in the local migratory practices. Winter loss data was not collected and falls outside the scope of the paper, because, as stated above, the fact that these colonies can survive mite infestations has already been published in peer-reviewed work multiple times and is not a subject for debate, nor an argument made by the data presented, it is taken as a well-studied fact. Please refer to the numerous studies cited in this work.

l. 194: I assume that all colonies that were sampled in April and June 2014 survived the winter 2013/2014. Hence, strictly speaking, all colonies were "surviving" colonies". If this is not the case, please specify and differentiate between truely surviving colonies and replaced colonies in both groups.

I think it is better to keep writing mite-surviving and mite-susceptible colonies because otherwise the impression is given that the other colonies have not survived - but they had, because they were sampled!? Or didn't they? Then please specify.

Please amend the terminology throughout the manuscript.

Response: Please refer to above comment, this has been addressed.

l. 202, test stat (X2) 15.30: In Table 1, the value 15.03 is given. Please correct. Please explain "(from qpcr negative results)" given in the Table.

Response: This has been amended. Prevalence taken from the absence or presence of virus in the qpcr analysis, binary presence absence tests were not performed, only the quantitative pcrs but this is sufficient to show the presence or absence of a virus.

l. 203, test stat (X2) 6.75: In Table 1, the value 4.61 is given. Please correct.

Response: This has been amended.

l. 203, p value 0.009: In Table 1, the value 0,03 is given. Please correct.

Response: This has been amended.

l. 204: Acc to MatMeth, for virus prevalence pooled samples were collected in April and June 2014. Please provide this information somewhere in the legend of the Fig and explain/justify why you combined the results from two sampling time points.

Response: This has been amended. The graph depicts prevalence for viruses sampled in June which are most relevant. As season did not affect most viruses, this was decided as the most concise way of conveying the important results. We have clarified the season in the figure caption.

l. 209, figure 4: It must be Fig. 3. Please correct.

Response: This has been amended.

l. 217: Please stick to one terminology! For instance, either write “mite-surviving colonies and mite-susceptible colonies” or “colonies from the mite-surviving population and the mite-susceptible population”, but don't switch between susceptible, mite-susceptible, treated etc. colonies.

This is especially relevant if you use a term here (treated) that cannot be found in the Figure or elsewhere.

Response: Changes have been made to reinforce “surviving” and “susceptible” as the central terminology, however, sometimes these terms are accompanied with “untreated” and “treated” to emphasize the fact that the colonies that have resistance to the parasites are not given any treatment for varroa. This is an important distinction and we press that it should remain.

l. 223: I cannot find results for individual adult workers in Table 1, only for (i) individual worker brood, (ii) pooled adult worker, and (iii) individual viral loads pupae: Likewise searching for X2 33.751 did not yield a result. Please correct.

Response: These data were analysed using a χ2 test and not a glm. Therefore table 1 should not have been referenced. The text has been amended in methods and results.

l. 225: Please be consistent: If the legend of the axis says "wandering mites" don't use "phoretic mites" in the figure legend.

Response: There is no mention of “wandering” mites in text or in figures or tables. Phoretic is used for mites on adult worker bees and the term “associated” is used for mites found in brood to indicate that the mite and brood samples were paired.

l. 238: For pupae, I can only find data for individual viral loads but not for prevalence in Table 1. Please correct.

Response: Models 3a-c display the data for individual pupae and mite sample prevalence. These models have been set lower in the table to reflect the order of presentation in text and are now models 5a-c.

l. 240: In Table 1 the term "viral load" is used, here it is "virus abundance". Please be consistent and rather take viral load throughout.

Response: Previous reviews had the term changed to “abundance”, we have now made this consistent throughout the manuscript and supporting documentation.

l. 246: Fig. 5 is about prevalence and abundance, not only prevalence. “Abundance” is better named viral load. Please correct.

Response: Figure 5 is strictly about abundance, the text has been corrected, please see above comment in regards to chosen terminology.

l. 246: Fig. 5 is supposed to show surviving pupae and associated mites (see l. 241), not adult workers and phoretic mites. Please correct.

Response: This has been amended.

l. 247: Fig 5 is a box plot showing more than just means. Please describe correctly what is shown and specify the box and error bars (SD? SEM?).

Response: This has been amended.

l. 249: Here the use of "surviving" colonies is especially problematic. Since these data are from autumn, the use of the term surviving implies that the data of surviving colonies from both groups (mite survivjng and mite susceptible) are shown here, but I assume that surviving actually means “mite surviving”. Please correct.

Response: This has been addressed in previous comments: Winter loss data was not collected and falls outside the scope of the paper, because, as stated above, the fact that these colonies can survive mite infestations has already been published in peer-reviewed work multiple times and is not a subject for debate, nor an argument made by the data presented.

l. 256: Please provide data on what happened over winter. Did the described differences had any influence on the mortality rate? Did the highly infested and virus infected colonies all die or did they survive? This information is lacking but highly relevant. Please add this information.

Response: This has been addressed in previous comments.

l. 259: Please also mention and properly discuss the possibility that the experimental design (pools of entire bees were analyzed, only DWV-A was analyzed instead of DWV-B or even better both variants) was not optimal for answering the question.

Response: This has been addressed in previous comments.

---

## [Editor Report · Decision Letter 3]

29 Jun 2023

PONE-D-22-33692R3Virus infections in honeybee colonies naturally surviving ectoparasitic mite vectorsPLOS ONE

Dear Dr. Oddie,

Thank you for submitting your manuscript to PLOS ONE. After careful consideration, we feel that it has merit but does not fully meet PLOS ONE’s publication criteria as it currently stands. Therefore, we invite you to submit a revised version of the manuscript that addresses the points raised during the review process.

We look forward to receiving your revised manuscript.

Kind regards,

Stephen J. Martin

Academic Editor

PLOS ONE

Journal Requirements:

Additional Editor Comments:

I can see that this ms has been reviewed by 4 people at various times and 3 suggest minor revisions. The final and latest one suggested major revision which I am satisfied you have addressed fully in your changes to the final versions and in the response letter. As the new editor for the ms I have carefully read the most recent version and suggest the following minor changes before final acceptance as I am not requesting any more reviews so these are the final changes.

Line 39 remove full stop after ‘many cases’

Line 81 reword (BQCV and LSV),

Line 104 Change 100 to ‘One hundred’

Line 272, maybe add 'as suggested by Grindrod and Martin (2021)'?.

Line 327 remove full stop before refs

Check refs for Latin names and put in italics.

Both these previous studies found lower DWV viral loads in resistant populations so worth adding into your discussion as it strengthens your findings:

de Souza FS, Allsopp M, Martin SJ (2020) Deformed wing virus prevalence and load in honeybees in South Africa. Arch Virol (166): 237–241.

Mendoza Y et al., (2020) Unravelling Honey Bee–Varroa destructor Interaction: Multiple Factors Involved in Differential Resistance between Two Uruguayan Populations. Vet Sci (7):116.

Also, Grindrod and Martin (2021) states in the discussion that “a reduced mite burden also reduces the number of viral vectors” and you come to the same conclusion so maybe worth citing this ref in that context to support your ideas.

Grindrod I, Martin SJ (2021) Parallel Evolution of Varroa Resistance in Honey Bees; a common mechanism across continents? Proc. R. Soc. B 288: 20211375.

---

## [Author Response · Author response to Decision Letter 3]

20 Jul 2023

Journal Requirements:

Author comment: To our knowledge, none of the references cited have been retracted, species names have been italicized as per the editor’s request.

Additional Editor Comments:

I can see that this ms has been reviewed by 4 people at various times and 3 suggest minor revisions. The final and latest one suggested major revision which I am satisfied you have addressed fully in your changes to the final versions and in the response letter. As the new editor for the ms I have carefully read the most recent version and suggest the following minor changes before final acceptance as I am not requesting any more reviews so these are the final changes.

Response: We are very happy with the editor reassignment and thank you very graciously for the feedback.

Line 39 remove full stop after ‘many cases’

Line 81 reword (BQCV and LSV),

Line 104 Change 100 to ‘One hundred’

Line 272, maybe add 'as suggested by Grindrod and Martin (2021)'?.

Line 327 remove full stop before refs

Check refs for Latin names and put in italics.

Both these previous studies found lower DWV viral loads in resistant populations so worth adding into your discussion as it strengthens your findings:

de Souza FS, Allsopp M, Martin SJ (2020) Deformed wing virus prevalence and load in honeybees in South Africa. Arch Virol (166): 237–241.

Mendoza Y et al., (2020) Unravelling Honey Bee–Varroa destructor Interaction: Multiple Factors Involved in Differential Resistance between Two Uruguayan Populations. Vet Sci (7):116.

Also, Grindrod and Martin (2021) states in the discussion that “a reduced mite burden also reduces the number of viral vectors” and you come to the same conclusion so maybe worth citing this ref in that context to support your ideas.

Grindrod I, Martin SJ (2021) Parallel Evolution of Varroa Resistance in Honey Bees; a common mechanism across continents? Proc. R. Soc. B 288: 20211375.

Response: All changes have been made precisely as the editor has specified, new references have been cited and discussed in text and all minor edits have been made (line numbers match those specified)

---

## [Editor Report · Decision Letter 4]

28 Jul 2023

Virus infections in honeybee colonies naturally surviving ectoparasitic mite vectors

PONE-D-22-33692R4

Dear Dr. Oddie,

We’re pleased to inform you that your manuscript has been judged scientifically suitable for publication and will be formally accepted for publication once it meets all outstanding technical requirements.

Kind regards,

Stephen J. Martin

Academic Editor

PLOS ONE

Additional Editor Comments (optional):

A nice study
---

## [Editor Report · Acceptance letter]

1 Sep 2023

PONE-D-22-33692R4 

Virus infections in honeybee colonies naturally surviving ectoparasitic mite vectors 

Dear Dr. Oddie:

I'm pleased to inform you that your manuscript has been deemed suitable for publication in PLOS ONE. Congratulations! Your manuscript is now with our production department. 

Kind regards, 

on behalf of

Prof. Stephen J. Martin 

Academic Editor

PLOS ONE